



# The Impact of a Solar Extreme Event on the Middle Atmosphere, a Case Study

Thomas Reddmann[1], Miriam Sinnhuber[1], Jan Maik Wissing[2], Olesya Yakovchuk[2], and Ilya Usoskin[3]

[1]Karlsruhe Institute of Technology, Germany
[2]University of Rostock, Germany
[3]Space Physics and Astronomy Research Unit and Sodankyla Geophysical Observatory, University of Oulu, Finland

**Correspondence:** Thomas Reddmann (thomas.reddmann@kit.edu)

**Abstract.** A possible impact of an extreme solar particle event (ESPE) on the middle atmosphere is studied for the present-day climate and geomagnetic conditions. We consider an ESPE with an occurrence probability of about 1 per millenium combined with an extreme geomagnetic storm (GMS) following the ESPE. The strongest known and best documented ESPE of 774/5 CE is taken as a reference example and established estimates of the corresponding ionization rates are applied. The

ionization rates due to the energetic particle precipitation (EPP) during an extreme geomagnetic storm are up-scaled from analyzed distributions of electron energy spectra of observed geomagnetic storms. The consecutive buildup of NOx and HOx by the ionization is modeled in the high top 3D chemistry circulation model KASIMA, using specified dynamics from ERA-Interim analyses up to the stratopause. A specific dynamical situation was chosen which includes an elevated stratosphere event during January and maximizes the vertical coupling between the Northern polar mesosphere-lower thermosphere region and

the stratosphere and therefore allows to estimate a maximum possible impact. The results show a strong enhancement of NOx which causes a substantial decrease of ozone in the mesosphere and stratosphere, and a significant decrease of total ozone in the Northern hemisphere in spring, enduring into the midlatitude summer after the event. The geomagnetic storm causes strong ozone reduction in the mesosphere but plays only a minor role for the reduction in total ozone. In the Southern hemisphere, the long-lived NOy in the polar stratosphere which is produced almost solely by the ESPE, is transported into the Antarctic

polar vortex where it experiences strong denitrification into the troposphere. For this special case, we estimate a $NO_3$ wash-out which could produce a measurable signal in ice cores. The reduction in total ozone causes an increase of the UV erythema dose of less than 5% which maximizes in spring for Northern latitudes of 30° and in summer for Northern latitudes of about 60°.

## 1 Introduction

Strong events of solar activity, such as the Carrington flare event (Carrington (1859)), have been observed now for more

than 150 years and since the satellite era also their energetic output is now well documented (see for example Aschwanden et al. (2017)). These events often produce a strong flux of energetic particles from the Sun which, when precipitating into the Earth's atmosphere, by ionization processes (Usoskin et al. (2011)), cause the formation of the radicals NOx and HOx. This results in an additional ozone destruction in the middle atmosphere (see for example Sinnhuber et al. (2012)). For solar eruptions observed during the last decades there exist many studies, observational as well as modelling ones, documenting this





solar-terrestrial interaction (see for example studies in the context of the WCRP SPARC High Energy Particle Precipitation in the Atmosphere (HEPPA) initiative as Funke et al. (2011a), Funke et al. (2017), Sinnhuber et al. (2022)). In addition, some studies indicate a possible small, but not conclusive surface impact (see for example Seppälä et al. (2009), Maliniemi et al. (2014), Calisto et al. (2011)). Even stronger events with a probability of $\sim 1/1000 \, \text{y}^{-1}$ have put marks in the concentration of cosmogenic isotopes in natural stratified records (e.g., tree trunks, ice cores) during the last millenia (Usoskin (2017), Cliver

et al. (2022)). The impact of such extreme events have been studied mostly in the historical context to simulate the production of isotopes in the lower stratosphere and their transport to the surface to compare with the isotopic records (eg. Sukhodolov et al. (2017)). Here we apply ionization rates estimated for an extreme solar event in our chemical model of the atmosphere to study its chemical impact and its consequences for the ozone layer under present-day conditions, i.e., if such an event occurred nowadays. The main objective of this study is to estimate the maximum direct impact which such an extraordinary event

may have on the chemical state of the atmosphere, and in consequence for the erythemal UV-dose on-ground. As the ozone destroying radicals are generated over a wide range of altitudes in the middle atmosphere, it is also the specific dynamical situation which determines its impact. We focus on the Northern hemisphere (NH) and apply a dynamical situation where the vertical coupling in the middle atmosphere is particularly efficient.

The paper is structured as follows: First, we construct an extreme solar scenario with properties of an extreme event observed

in the past or estimated to have an occurrence rate of about 1/1000 y and estimate the corresponding ionization rate. Then we apply this scenario to the specific atmospheric situation, calculate the additional NOx and its impact on ozone. Finally we estimate the consequences for the additional UV dose following the event for NH mid latitudes.

## 2 The extreme scenario setup

A solar eruption event has several components: the very first signal of such an event detectable on Earth is a flare of electro-

magnetic radiation emitted in the eruption at the solar surface. Observable from $\gamma$-ray energies down to the visual spectrum (white flare), the total impact of the flare on the Earth's middle atmosphere is small and negligible in terms of the chemical impact (Pettit et al. (2018)) and therefore for the possible dynamical coupling. Then, particles accelerated in this explosion and farther in the corona and interplanetary medium reach the Earth within minutes to hours after the flare, the so called solar proton event (SPE). Here the highest energies of the particles are observed up to several GeV which initiate a nucleonic cascade

in the atmosphere. This can reach even the ground and, when colliding with atmosphere's main constituents, cause nuclear reactions which can form, in particular, the cosmogenic isotopes $^{14}$C or $^{10}$Be. As one of the strongest of such events detected in paleo-nuclide records so far, the possible solar eruption dated at 774/5 CE has been studied extensively (see eg. Usoskin (2017)) including its impact on the atmosphere (Sukhodolov et al. (2017)). Compared to the the strongest directly observed SPE in 1956, Mekhaldi et al. 2015 estimated a factor 40 higher fluence to produce the observed nuclide concentrations. More

recent studies hint to a even stronger fluence. Cliver et al. (2022) estimate a 70x particle fluence compared to the 1956 event and we scale the ionization production rates of the 1956 event accordingly.





In addition, a strong eruption on the Sun's surface is often accompanied by a so-called coronal mass ejection which when directed to Earth hits the magnetosphere and causes a major geomagnetic storm (GMS), typically in the range of two days after the flare event. The energies of the particles of the GMS (mostly electrons) are in the range of a few keV to about 1 MeV,

and they impact mainly the mesosphere and lower thermosphere. The strength of the disturbance of the geomagnetic field can be expressed by planetary indices like Kp and Ap but their relation to ionization rates for extreme cases could be highly non-linear. Here we take a different approach and assume the results of the study of Meredith et al. (2016) to be applicable for strong geomagnetic events when studying the interaction with the atmosphere. Meredith et al. (2016) conducted an extreme values analysis of the electron flux measured in the MEPED (Medium Energy Proton and Electron Detector) $90°$-pitch-angle

detector for the three energy intervals provided by the instrument. We assume here that we can apply these results also for particles entering the atmosphere ($0°$ pitch-angle) and scale observed strong GMS events according their distribution. We use their list of extreme events at 30 keV and 300 keV and take the strongest events as representatives of extreme geomagnetic events. Note that their list excludes SPEs. The 2003-Nov-20 event is the highest at $L_* = 4.5$ and third in rank at $L_* = 6$ for 30 keV, the 2010-Apr-6 09-12UT event is highest at $L_* = 6$ and the several 2010-Apr-6 events are together highest also at

$L_* = 4.5$ for 300 keV. The electron fluxes for these events have about a 1 in 10 year probability when inspecting their Fig. 6. From this Figure, one can also deduce enhancement factors for 1-in-100 year events which are about 2x for low and mid energy electrons, and 10x for high energy electrons compared to the 1-in-10 year events. Our extreme GSM event is finally constructed in the following way: We take the ionization rates (including protons) calculated with the AIMOS-AISstorm 2.0 model (described in Nesse Tyssøy et al. (2022)) interpolated to our model grid for 2003-Nov-20 and 2010-Apr-06, scale them

with factors 2 and 10, respectively, and add them up as one event with a duration of one day. As the AIMOS-AIStorm model includes electrons limited to $< 300$ keV, the lower boundary of a realistic electron ionization production rate is about 70 km.

Finally, we assume that SPE and GSM indeed can be combined for this sensitivity study, start with a one-day extreme SPE and after two days with the extreme GSM. Figure 1 shows the three days time and zonal average ionization production rate. We note that this approach may be not physically completely consistent but should yield a meaningful upper value for the strength

of an extreme event.

The impact of the solar events on ozone is mainly given by the particle induced buildup of NOx which catalytically reacts with ozone. NOy, which includes all nitrogen containing species except $N_2O$ including the reservoir gases for NOx, has a very long lifetime in the stratosphere where it contributes effectively to ozone destruction. The impact on ozone is the key parameter for any dynamical atmospheric impact of such events, as ozone is the most important radiatively active gas for solar

radiation and changes in ozone will change the temperature distribution the middle atmosphere, and winds and propagating waves as a result. In addition, as the ozone density maximizes in the stratosphere, any reduction of ozone will also change the UV intensity at the surface. The SPE induced ionization rate maximizes in the upper stratosphere and lower mesosphere, the GMS has its maximum ionization in the lower thermosphere. In order to estimate a possible maximum impact of a solar event, a dynamical situation is appropriate where the portion of NOy built up in the mesosphere/lower thermosphere (MLT) during

the event survives photochemical destruction and is transported into the lower stratosphere, where its lifetime is of the order of many years. This sets the date of such an experiment to the winter season.

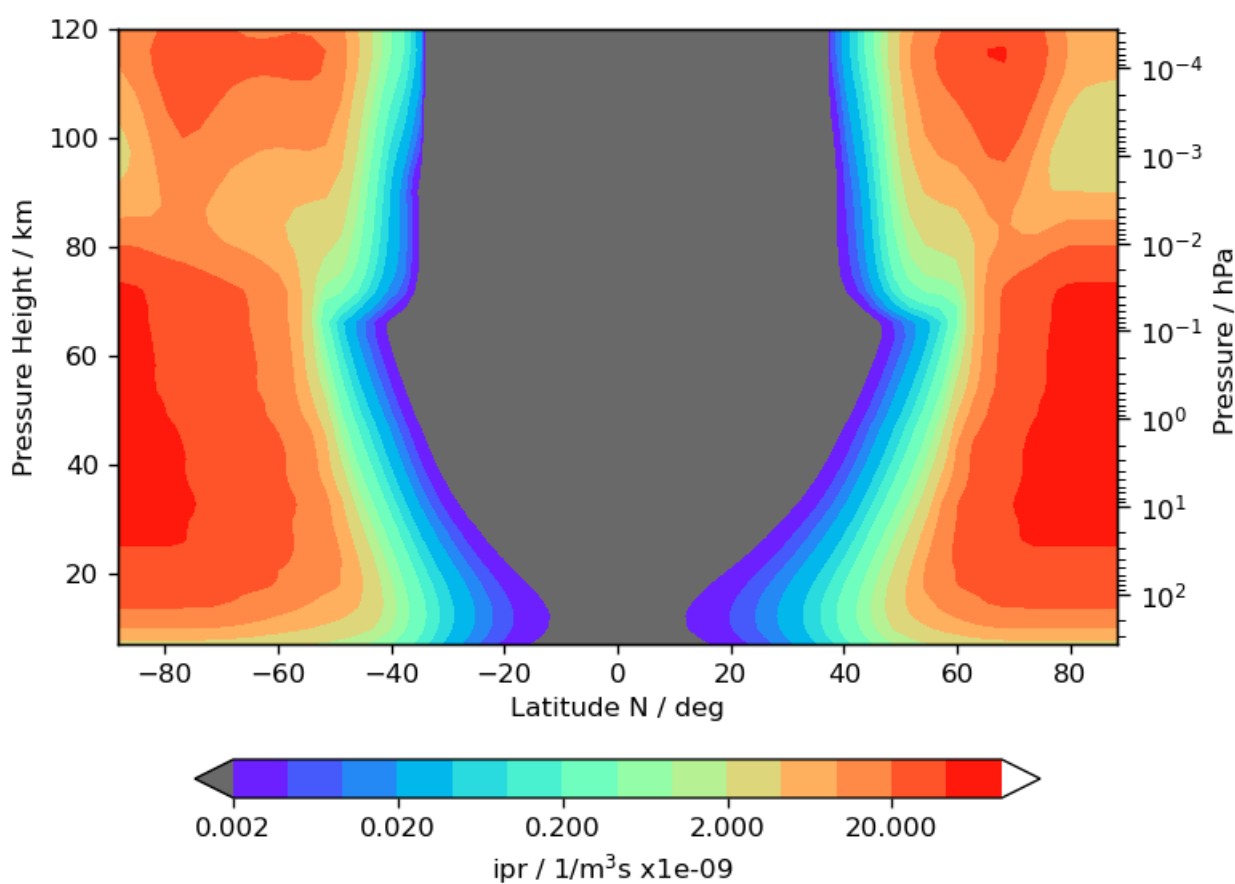

**Figure 1.** The ionization production rate of the extreme event (mean of January 23 - 25 2009).

In satellite observations from instruments like MIPAS on ENVISAT (Fischer et al. (2008)) strong NOy intrusions from the lower thermosphere into the mesosphere and upper stratosphere have been observed after sudden stratospheric warmings (SSWs) when accompanied by so-called elevated stratosphere events. These show a strong downward transport of air from the MLT (Holt et al. (2013)). For the Northern hemisphere it has been shown that these events also yield the strongest ozone impact (Sinnhuber et al. (2018)). In order to test if an extreme geomagnetic storm can reach a similar impact as the SPE which maximizes lower in the mesosphere, we synchronize the SSW and the geomagnetic storm. As an example for such an dynamical situation, we use the SSW of Jan 21 2009 which has also been studied in the HEPPA-II experiment (Funke et al. (2017)).



In summary, the extreme event we study is a combination of a SPE and a GMS. The ionization production rate is derived from observed events which have been scaled to a strength estimated for an event with a occurrence rate of about 1 in 100 - 1000 years, synchronized with an elevated stratosphere event.

## 3    Description of the model and experiment

For the purpose of this sensitivity study we use the KArlsruhe SImulation Model of the middle Atmopshere (Kouker et al.
(1999)) in the version described in Sinnhuber et al. (2022). The model solves the basic equations in spectral form in the altitude range between $300\,hPa$ and $3.6 \times 10^{-5}\,hPa$ with the pressure height $z = H \log(p/p_0)$ ($H = 7\,km$ and $p_0 = 1013.25\,hPa$) as vertical coordinate. It uses radiative forcing terms for UV-Vis and IR, and a gravity wave drag scheme. In order to yield a realistic meteorology, the model is relaxed (nudged) to ERA-Interim meteorological analyses (Dee et al. (2011)) between the lower boundary of the model and $1\,hPa$. A full stratospheric chemistry including heterogeneous processes is adapted to include
source terms related to particle ionization. For the production of HOx the parameterization of Solomon et al. (1981) is used. For the production of NOx, 0.7 NO molecules are produced per ion pair and 0.55 N atoms in ground state. The model also includes a $HNO_3$ production from proton hydrates based on the parameterization of de Zafra and Smyshlyaev (2001) which has been modified to be dependent on actual ionization rates. The model participated in all the three HEPPA inter-comparisons (Funke et al. (2011b), Funke et al. (2017), Sinnhuber et al. (2022)). The model has proven to simulate a realistic chemistry in
the middle atmosphere for NOy intrusions which is sufficient to study the direct impact.

    The following simulations are performed: one with the extreme event (EXT), where the ionization rates from the extreme event are applied from January-21 to January-23., and one with a background ionization only (REF). The background ionization rate is a mean from AIMOS ionization rates (Wissing and Kallenrode (2009)) for minimum Ap indices. In addition, a simulation which only includes the solar proton event is used to compare the contributions of the two components.

## 4    Results

### 4.1    The NOy intrusion

Figure 2 shows the mean of NOy ($= NOx + HNO_3 + ClONO_2 + BrONO_2 + HONO + N_2O_5 + BrONO$) (gas phase only) where $NOx = N + NO + NO_2$, over the Northern polar cap (latitude > 70°N) for a year with the dynamics and chemical boundary conditions of 2009, but with the extreme solar event described in Section 2 included. The two components of the event (SPE
and GMS) can clearly be seen as an instantaneous increase over all altitudes and a tongue of increased NOy values transported from the MLT within the elevated stratopause event, respectively. In the specific dynamical situation, the downward transport of the GSM component from the MLT is fast enough to reach the photolytic 'safe' stratosphere at the end of the NH winter which then causes a significant NOy enhancement in the upper stratosphere. It stays there and is diluted during the summer, and is then transported further down in the beginning of the winter season. The enhancement related to the SPE is seen down to
the tropopause and stays there over the whole year with relative enhancement of more than 20%. NOx is lost through transport





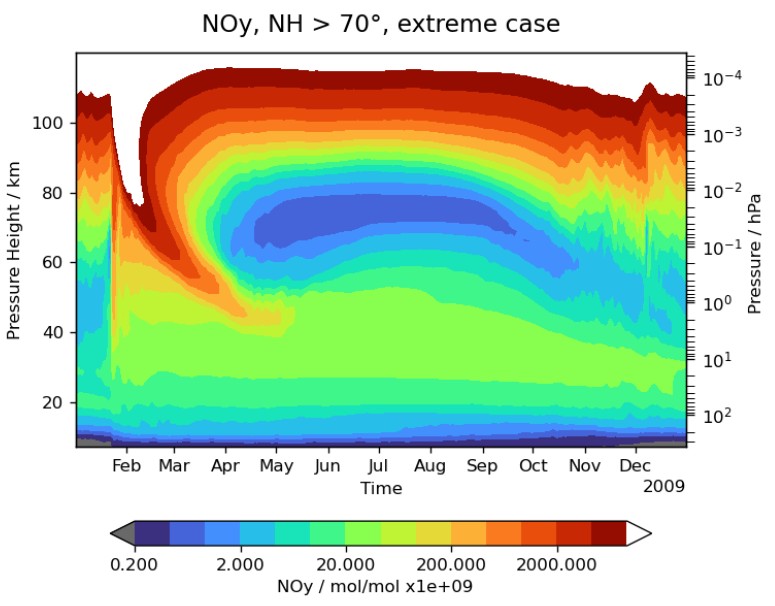

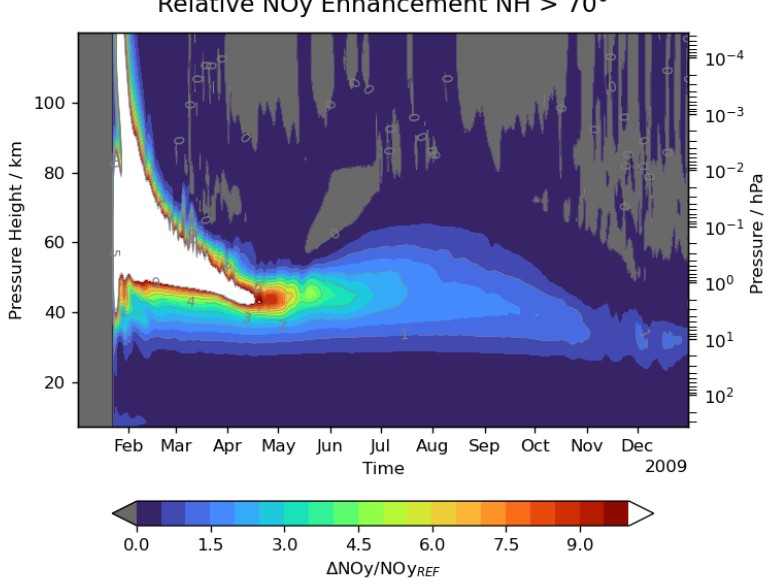

**Figure 2.** Top: NOy mixing ratio in the course of one year with specified dynamics of the year 2009 (ERA-Interim) and applying the extreme scenario case as described in the text. For the polar cap the intrusion connected to the MLT part dominates around the stratopause, below NOy stems from the SPE. Bottom: Relative enhancement of NOy compared to reference run.





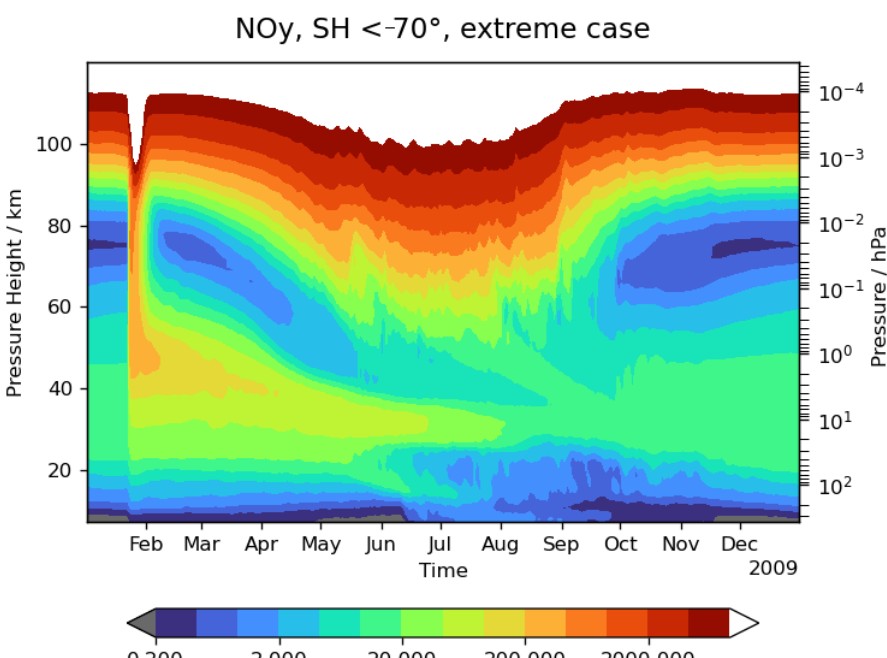

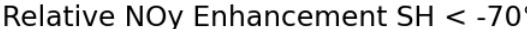

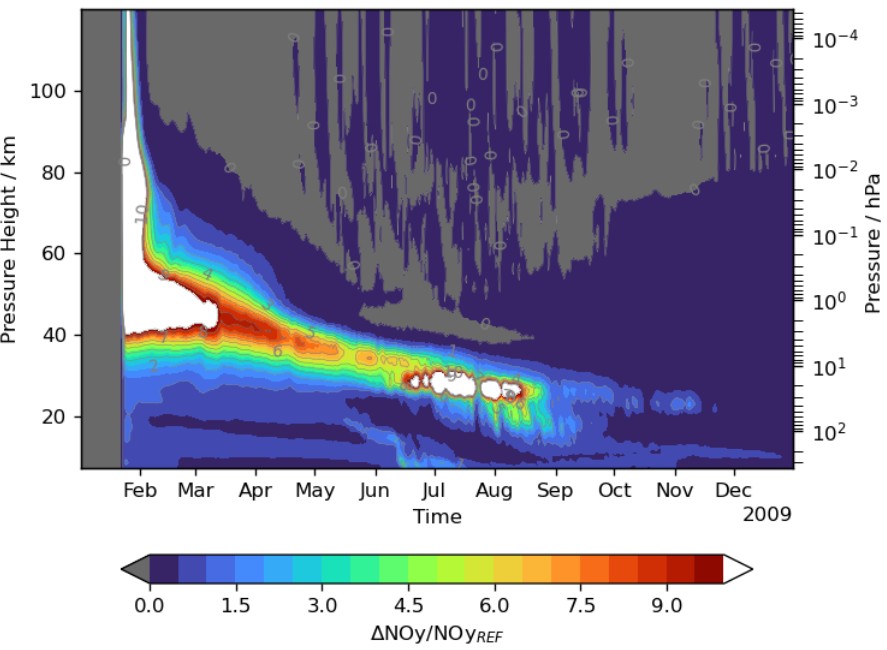

**Figure 3.** As in 2, but for 70S

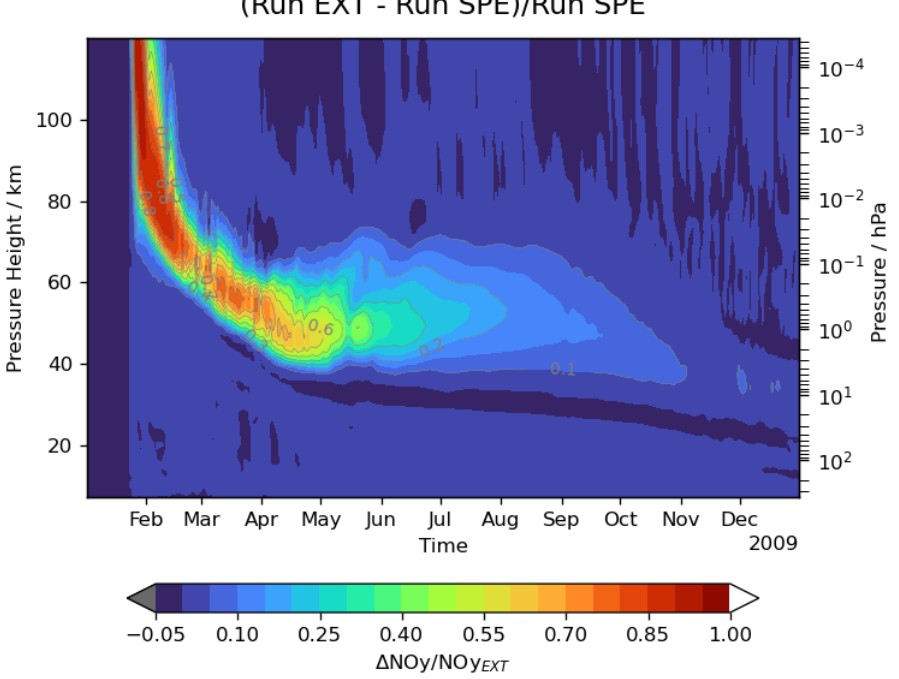

**Figure 4.** The relative contribution of the GSM component to the NOy intrusion (latitude > 70N), calculated as the difference between the EXT run and a run with the SPE component only, relative to the full event.

into the troposphere where the main reservoir in the lowest stratosphere $HNO_3$ is washed out, or into the mesosphere where it is photolytically destroyed.

In the Southern hemisphere (shown in Figure 3), the NOy enhancement is essentially from the SPE, as the MLT component is photolytically destroyed during the summer season. We also observe a higher relative enhancement here, as the excess NOy

in the upper stratosphere is transported downward in the beginning winter, compared to the upward draft in the NH summer. In addition, de- and renitrification by sedimentation of condensed $HNO_3$ can be observed, resulting in an effective removal of nitrate from the stratosphere into the troposphere. The relative maxima in the south winter middle stratosphere are related to the higher supply of $HNO_3$ where in the reference run this air mass already was depleted.

The contribution of the GMS to the NOy at high Northern latitudes (>70°N) in the intrusion is shown in Figure 4. It is limited

in altitude to above about 40 km and is diluted to values below 0.1 in autumn.

The distribution of NOy over latitudes at 37 km altitude is shown in Figure 5. At this altitude the contribution of NOy to ozone depletion maximizes (see for example Brasseur and Solomon (2005), their Figure 6.1). A closer inspection shows that at this altitude also NOy from the NH is transported to the SH and significantly contributes to the NOy after some months, as part of the diabatic circulation from the summer to the winter pole.

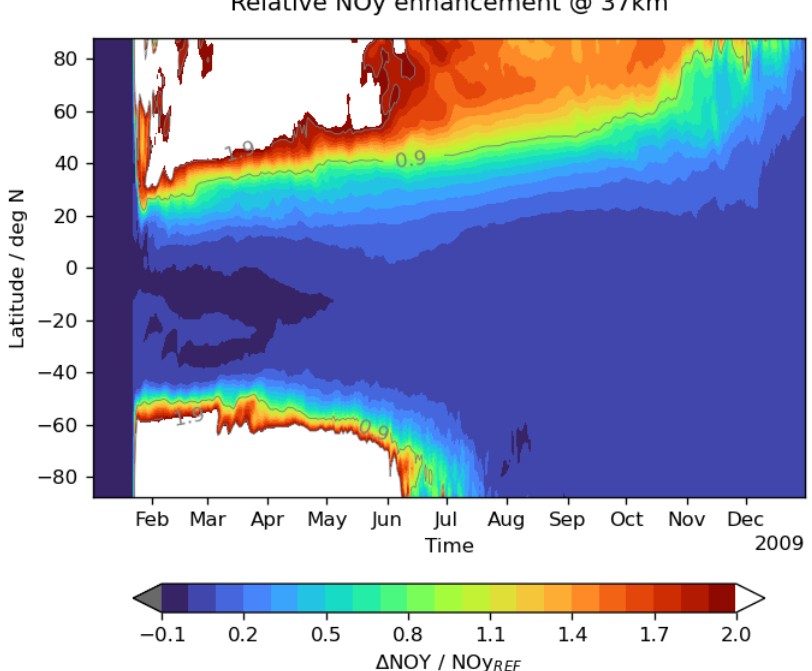

**Figure 5.** Relative enhancement of NOy by the the extreme event at 37 km pressure height.

## 4.2 The Impact on the Global NOy Budget

To assess the impact of the NOy intrusions on the global NOy budget, we first calculated the total column of NOy with and without the event, as shown in Figure 6. The two hemispheres show remarkable differences, with a higher total relative NOy enhancement at high Southern latitudes in the first six months but a fast decay after, and a longer lasting NOy enhancement in mid and high latitudes in the Northern hemisphere (see also 7). From Figure 3 which shows mixing ratios, one can deduce that the strong relative enhancement is affected by differences in the depletion of gaseous NOy by denitrification in the reference and extreme run. This gets clearer when showing the absolute change of the total columnar content. The change of total amount NOy (in units of mol) for different latitude bands by the event is shown in Figure 7. In the Southern hemisphere, the fast decay shortly after the event connected to the photolytical destruction in the polar mesosphere, then some stabilisation in the winter followed by the denitrification in the cold polar stratosphere can clearly observed. The Northern hemisphere is characterized by a more continuous decay, stabilizing in the next winter season.

## 4.3 Impact on ozone

The NOy enhancement causes additional ozone depletion via the catalytic reaction of NOx with $O_3$. This is shown in Figure 8 as the relative change of ozone. In the MLT, there is also a short lived ozone enhancement which is connected to the dissociation



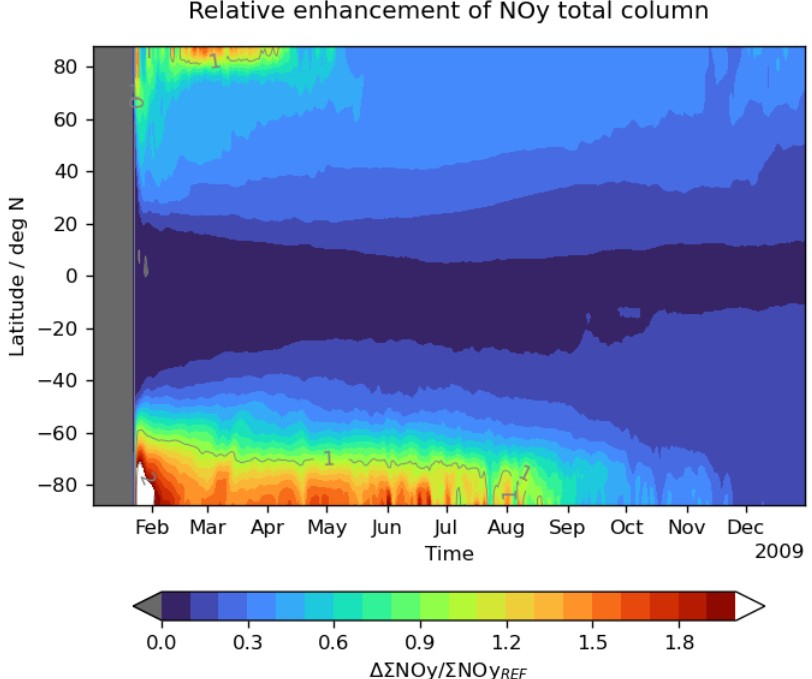

**Figure 6.** Enhancement of the total NOy column (> 7 km pressure height) relative to the reference run.

of $O_2$ by atomic nitrogen. In addition, the ozone decrease in the stratosphere causes less absorption of solar UV and a small
build-up of ozone below the depleted zone. Mirroring the different amount of NOy transported into the stratosphere, the
additional ozone loss is higher in the Southern hemisphere, whereas the ozone loss in the Northern hemisphere is lasting for a
longer time. This just reflects the loss of NOy in the Southern hemisphere in the course of the polar winter.

The change of the ozone column relative to the reference run is shown in Figure 9. It shows the already described asymmetry
in the two hemispheres, with exceptional high changes of total ozone loss in the Antarctic vortex. Total ozone recovers to nearly
normal values at the end of the simulation year in the Southern hemisphere. On the other hand, in the Northern Hemisphere
the maximum loss in midlatitudes is less than 5% but a deficit of total ozone of a few percent stays permanently throughout the
total year. The contribution of the GSM to the loss in total ozone is negligible (not shown).

## 5 Discussion

### 5.1 The specific dynamical situation and the total NOy input

The simulation presented is based on a specific dynamical situation which was chosen to maximize the impact of an solar
extreme event in the Northern hemisphere. The results show that despite a kind of optimum situation for the indirect NOy





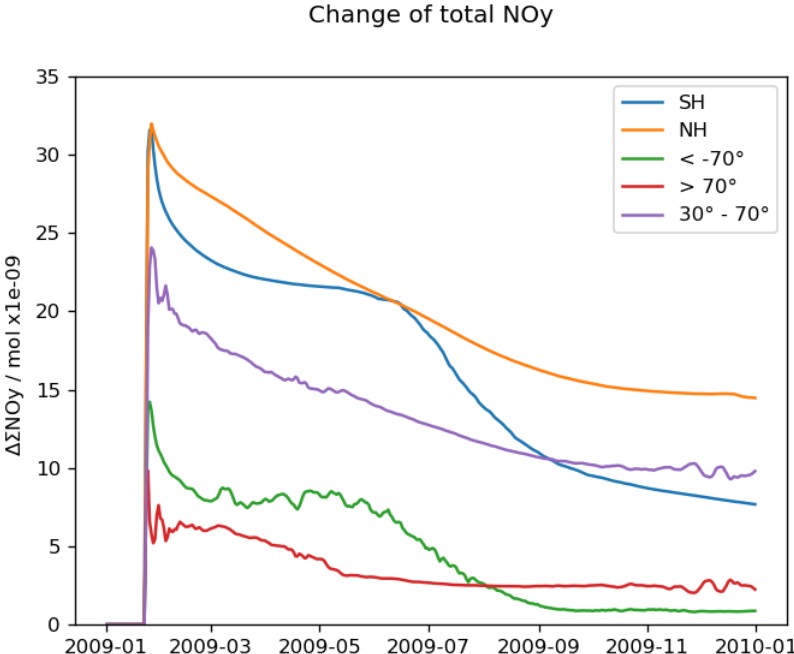

**Figure 7.** The NOy signal from the extreme event (EXT- REF) for different latitude bands as the total amount of NOy in mol.

effect via NOy transported downward from the MLT, the contribution of the GSM to NOy is of minor importance compared to the impact of the SPE. For comparison, we tested also other dynamical situations, from early to late winter (only in the winter season NOy is stable in the mesosphere): in winter 2009/2010 but the event put on doy 300 in 2009, a rather early and strong

mid-winter warming (December 1998), and the SSW in mid-January 2004, which from the date is quite similar to the set-up studied. For the EXT scenario, we estimate the initial total NOy input to about 65 Gmol which decreases to about 25 Gmol after one year (see Figure 7). The undisturbed total NOy amount in the atmosphere prior to the event above about 7 km altitude is simulated to 135 Gmol. The experiments for 2004 and 1998 yield the same initial amount of the intrusion, but decay to about 10 Gmol and 25 Gmol after one year, respectively. The injection in 2009 doy 300 yielded with 53 Gmol slightly less initial

NOy and showed a rather fast decay to only a few Gmol after one year.

From these comparisons, the scenario selected seems to be an example of a strong and long enduring impact of the event. Obviously not just the GSM part profits from the dynamical situation but also the mesospheric part of the SPE. To estimate the range of the impact of strong solar events under different dynamical situations needs more simulations, also including feedback effects on the dynamics. This is beyond this study and will be addressed in a follow-up paper.

The total NOy mass injected in experiment EXT of about 65 Gmol corresponding to about 40% of the total atmospheric NOy content is substantially greater than what has been found in observation during the last decades. For example, Vitt and Jackman (1996) estimate a few percent increase of NOy after strong SPEs in the Northern polar stratosphere, Funke et al. (2014)







**Figure 8.** Ozone change in the extreme run relative to the reference run ((REF-EXT)/REF) related to the NOy intrusion through the extreme event (EXT - REF) for the Northern and Southern polar cap (latitude $|\phi| \geq 70°$).



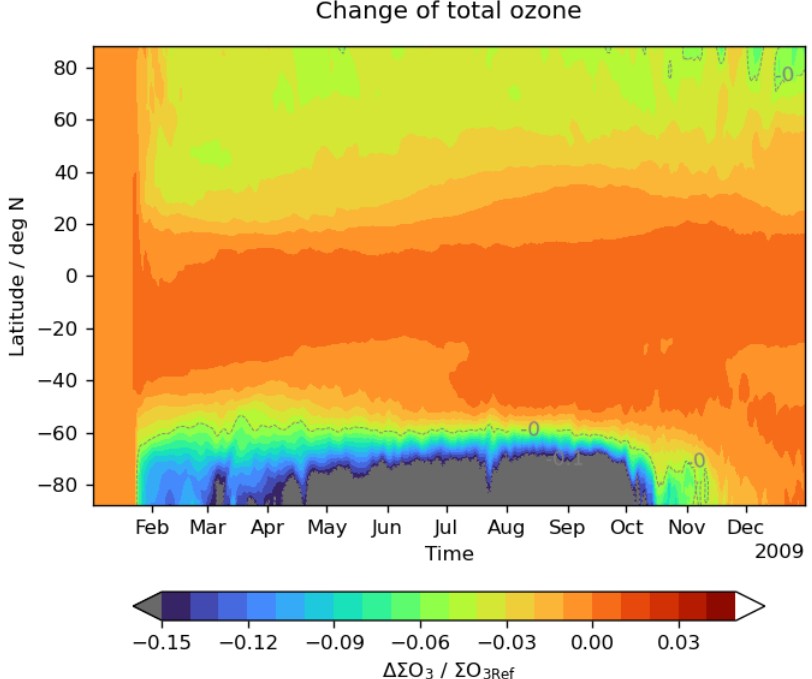

**Figure 9.** The change of total ozone caused by the solar extreme event relative to the reference run.

estimated the total NOy input from satellite observations for the period 2002 to 2012 to a few Gmol per year, Reddmann et al. (2010) deduced the total NOy input from a model simulation for the period 2003 to 2004 to about 2 Gmol. In terms of NOy
input, the NOy buildup by the solar event studied is about 10 to 30 times higher than what is estimated for solar particle events observed in the satellite era which is of the order of the scaling factor we apply to the ionization rates of the SPE. In this sense, the solar extreme event is also an atmospheric extreme.

   The budget of NOy in the atmosphere on the longer term depends on the source strength of the event and the loss terms by photochemistry and exchange processes to the troposphere where NOy is removed by precipitation.

For the source strength we use ionization rates estimated for the event in 774 CE. There are several steps to estimate ionization rates for such events which all have factors of uncertainty. This is extensively discussed in Usoskin et al. (2011) and also in Sukhodolov et al. (2017). One of the main uncertainties of the SPE part is the energy spectrum of the protons which has been scaled from an event in 1956 with a rather hard spectrum, compared to other events in the satellite era Usoskin et al. (2020). A softer spectrum would shift the maximum NOy production to higher altitudes where the lifetime of NOy is reduced.
Therefore, the initial NOy production by the ionization we estimate is probably on the upper limit of what can be expected.

   The GSM contribution has been estimated from events observed in the past, but data at high flux levels are rare and have therefore substantial uncertainty. On the other hand, theoretical considerations (Vasyliunas (2011)) set an upper limit to the strength of a GSM in terms of Dst to about 2500 nT, which is probably outside the estimations we used. In addition, the





geographical distribution of the auroral oval depends on the strengths of the event. Here we used the distributions from the
AISstorm model for observed events and therefore miss the shift of the auroral oval to lower latitudes when the GSM strength
is much higher. The scaling factors taken from Meredith et al. (2016) have been derived for a particle population that is partly
trapped in the magnetosphere. Applying them to particle precipitation includes another source of uncertainty. But even a factor
of two higher contribution from the GSM part would not bring this component above the SPE contribution.

For the efficiency of the NO production we use in the model the results of Porter et al. (1976). Some studies indicate a
dependency of these factors on the ionization rate itself (Nieder et al. (2014)), with a kind of saturation effect, especially in the
thermosphere. This can be a major uncertainty for the production term and needs detailed ion chemistry studies.

Besides the source strength, the downward transport from the source region in the mesosphere to the stratosphere is essential.
From the studies within the SPARC HEPPA initative (Funke et al. (2011a), Funke et al. (2017)) we know that many models do
not correctly simulate this transport in the polar winter. The KASIMA model we use here in combination with ERA-Interim
analyses yield a downward transport from the MLT which is general in good agreement with observations, also for a situation
with an elevated stratosphere as we use here. Finally, the atmospheric lifetime of NOy is determined also by the correct pattern
and strength of the Brewer-Dobson circulation. It determines the upward transport into the region with photolytic loss, and the
exchange to the troposphere. Mean age of air simulations with KASIMA yield here also a good agreement with observations
(Stiller et al. (2008), Haenel et al. (2015)).

In conclusion, the main uncertainties for the strength of the NOy intrusion for a selected dynamical scenario as in our
simulation is the energy spectrum of the particle flux and the initial ion chemistry production terms.

### 5.2 The NOy loss through the Antarctic winter

The loss of NOy in the Southern polar winter atmosphere deserves special attention. The KASIMA model includes heteroge-
neous processes in the polar winter stratosphere causing dentrification and renitrification of the cold air masses in the Antarctic
polar vortex through the nitric acid trihydrate (NAT) condensation of $HNO_3$ on water ice and the adsorption on sulfate aerosols,
followed by sedimentation of the larger particles into lower layers as part of the annual cycle. In the KASIMA model, this pro-
cess seems to be saturated: in the core of the Antarctic polar vortex denitrification is complete. Additional NOy brought into
the lower stratosphere by the intrusion does not change that but by the changed saturation pressure, more volume will take part
in the denitrification. The change of total NOy for the Southern polar cap in Figure 7 supports that finding: at the end of the
winter, most of the additional NOy of about 6 Gmol in the Southern polar cap is lost to the troposphere.

Inspection of Figure 3 shows that the major NOy renitrification in the Southern polar lower stratosphere takes place between
end of July to mid of August. NOy here is mainly in the form of $HNO_3$. Once transported to the troposphere, NOy is deposited
onto the surface by rain or snow within a few days. If we assume that the NOy is deposited in the form of $NO_3$ on the
polar cap, we deduce an average deposition value of $2.4 \cdot 10^{-6} \mathrm{g/cm^2}$. Using the approximation that the column integrated
particle produced NOy is deposited, Melott et al. (2016) estimate a deposition of about $1.4 \cdot 10^{-7} \mathrm{g/cm^2}$ for the 1956 event,
corresponding to $1 \cdot 10^{-5} \mathrm{g/cm^2}$ for the extreme event just by applying the scaling factor. As in our model loss processes
as photolysis and transport to lower latitudes is included, our lower value is in line with their estimation. With an average



precipitation of 166 mm/year for Antarctica and assuming that the nitrate is deposited within one month we estimate a mass mixing ratio for nitrate of $190\,\mathrm{ng/g}$ in a monthly layer of (evenly distributed) deposited snow. A typical detection limit for

nitrate events measured in ice cores with monthly to sub-annual resolution is of the order of $100\,\mathrm{ng/g}$ (Smart et al. (2014)), so we would expect that a nitrate signal could be observable for such an event in Antarctica. Note, that our model does not include tropospheric meteorology, so an analysis of regional precipitation patterns is outside the scope of this paper. Sukhodolov et al. (2017) estimate from their analysis that a nitrate signal would not be detectable for a Greenland ice core. This is essentially in agreement with our findings, which show a much smaller change of total NOy in the Northern polar cap in early spring. Here,

the higher polar temperatures and less stable polar vortex do not allow strong and fast denitrification, the primary cause of the high NOy depletion in the Southern polar lower stratosphere. In addition, we use analyzed winds and temperatures near to a realistic meteorology, whereas free running models may suffer in several aspects to correctly describe the polar dynamical and chemical processes in the lower stratosphere. For the case of the 774/5 event, ice core data for Antarctica do not show nitrate enhancements (Sukhodolov et al. (2017); Jong et al. (2022)). From Fig. 3 one may note that a strong denitrification signal is

only expected when the NOy tongue reaches the lower stratosphere when there are also low temperatures. A mismatch, for example an event occurring in Antarctic spring, would presumably result in a much broader, and therefore undetectable nitrate peak.

### 5.3 Increase of UV-Index by the additional ozone depletion

In order to estimate any health related impact by the reduced total ozone column, we calculate the corresponding change in the

UV index. The UV-Index (UVI) is calculated from the spectrally weighted and integrated solar irradiance where the weight function takes the erythemal action spectrum into account (CIE International Commission on Illumination (2019)). Here we use the simplified formula of Allaart et al. (2004), their equation (8) together with equation (6), to estimate the increase of UVI connected to the decrease of the total ozone column ($\Sigma O_3$) after a solar extreme event. In this formula, besides the dependence on the extraterrestial flux and the solar zenith angle, for cloud-free situations UVI depends on $\Sigma O_3$ only. For the column down

to 7 km, the lower boundary of the model, we get an increase of the UVI for noon as shown in Fig. 10. The absolute highest change of about 0.5 UVI occurs in the months March and April at latitudes of 25° North which include the region on Earth with the highest population density. The relative change of UVI reaches about 4 - 5% in the northern midlatitudes where also the summer months are affected. In order to set this change in a context, we remark that the ozone depletion by CFCs reached a maximum reduction of total ozone of about 3% in northern midlatitudes in 1995 (Weber et al. (2022)), quite similar to the

values we found for the extreme event (see Fig. 9), but lasting for more than a decade.

There is also a significant increase of UVI at high Southern latitudes, but not at latitudes which have a high population density.





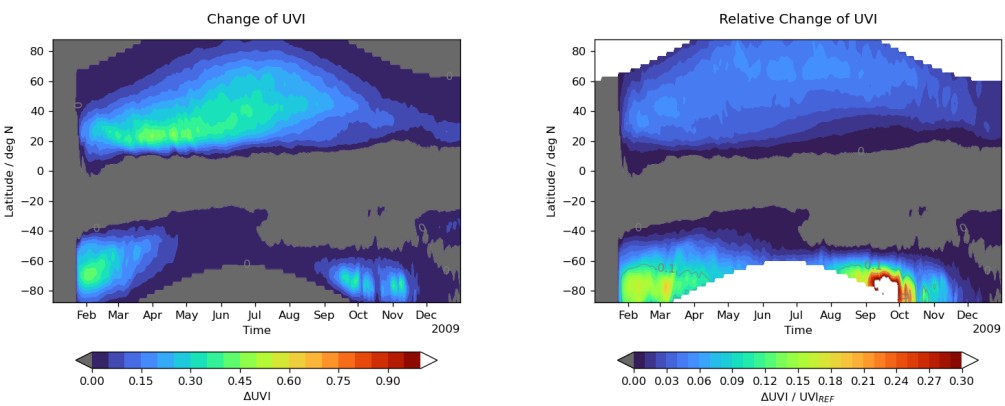

**Figure 10.** Change of the UV index by the extreme event for noon and the correspondent relative change of the UV index.

## 6 Conclusions

The purpose of this study was to analyze direct effects of an extreme solar eruption on the chemical state of the middle
atmosphere. We considered an extreme solar event with a strong SPE connected also with a strong GMS. The setup of the
numerical experiment was chosen to yield a maximum impact in the Northern Hemisphere. This was achieved by setting the
event to Northern Hemisphere winter in combination with an elevated stratosphere SSW event. The main findings are that
the GMS component from energetic electrons even in this dynamical situation leads only to a minor ozone reduction. The
ozone reduction in our extreme scenario stems essentially from the SPE and survives through the following boreal summer.
The studied event yields a significant enhancement of the UV index in populated latitudes in the year following the event.
The SPE caused a strong impact in terms of additional NOy and ozone loss in southern high latitudes, despite the event had
been put to Northern Hemisphere winter to minimize photochemical loss of NOy. This is related to the deep penetration of the
energetic particles at altitudes, where the loss of NOy is negligible, and the early on-set of the downward transport with the
beginning of the Antarctic winter in the middle atmosphere. Besides the related ozone loss, the denitrification of NOy in the
lower Antarctic stratosphere and the following washout in the troposphere can yield potentially measurable $NO_3$ signals in ice
cores. This, however, has not been observed so far, perhaps because the signal strength strongly depends on the timing of the
event. Therefore, nitrate signals in ice cores are not reliable indicators for solar activity.

*Author contributions.* TR prepared the experiments and wrote the paper, IU, JMW and OY provided ionization rates, MS critically discussed
and improved the manuscript

*Competing interests.* The authors declare that no competing interests are present.





*Acknowledgements.* This work was supported by the German Federal Ministry of Education and Research within the research program ROMIC-2 and by the Academy of Finland (projects ESPERA, No. 321882).

Model data used for this analysis are accessible through https://dx.doi.org/10.35097/917.





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
