# Peer review of "The Impact of a Solar Extreme Event on the Middle Atmosphere, a Case Study"

_Atmospheric Chemistry and Physics, 2023_

## Referee Comment (RC2)

**Review of the manuscript "*The Impact of a Solar Extreme Event on the Middle Atmosphere, a Case Study*" submitted by Reddmann et al. for publication in ACP**

Reddmann et al. describe results from simulations with the KASIMA 3D atmospheric circulation and chemistry model for the impact that an extreme solar particle event may have on atmospheric chemistry and surface UV radiation under present day conditions.

The model is forced by ionization rates estimated for an extreme event that has likely occurred in 774/5 CE. Tropospheric and stratospheric circulation of the KASIMA model are nudged to a specific meteorological situation from boreal winter 2008/2009 which includes a sudden stratospheric warming event that is considered to favor a strong impact of the particle event.

The topic is well suited for publication in ACP, the paper is in general well written, the methodology and the explanation of atmospheric chemistry responses seem sound to me. However, I think what the paper crucially lacks to be considered for publication is that the simulation results are put into context a) of an earlier simulation of such an extreme event, b) of the response to weaker, frequently simulated solar particle events, and c) of the role of the atmospheric background conditions.

a) There is an earlier simulation of impacts of the same 774/5 CE event by Sukhodolov et al. (2017) that is mentioned several times in the manuscript. However, I think the manuscript requires a very clear discussion of the question what we learn from the new study that we haven't learned from Sukhodolov et al. (2017). I understand that the new manuscript focusses much more on atmospheric chemistry than the earlier study, however, also Sukhodolov et al. show the response of $NO_x$ and $O_3$ for a period of half a year after the main event. Why not even put a figure that compares the two results with the same color scale and discuss the differences? Also Sukhodolov et al. briefly mention the likely minor impact on surface UV, admittedly using a different metric, but also here a discussion of similarity and differences would be very useful. Sukhodolov et al. go beyond the current study in the sense that they are not nudging to observations but run their model freely such that also the circulation responses can be analyzed. I think that this paper needs a short discussion of possible feedbacks from suppressed circulation and temperature responses on the chemistry. I know that the authors state this "needs more simulations … and will be addressed in a follow-up paper", but I guess there is experience from earlier studies which allows to estimate the potential importance of this.

b) The authors mention several other studies where effects of contemporary particle events have been observed and simulated. However, I don't think they sufficiently discuss differences between weak and strong particle events. For all their discussion of the response of atmospheric chemistry I would like to know if these responses are typical for particle events and just stronger because the ionization rates are larger. There is some discussion of this in Section 5.1 where for the total extra $NO_y$ it is said that it approximately scales to contemporary events as the ionization rates do. I would consider this the central result of the study. At least for total $NO_y$, we can estimate the

effect of events by simply scaling simulation results by the ratio of ionization results. Very useful, because we may not need new simulations for new events. Is this true beyond just total $NO_y$ increase? Are there non-linearities in other parameters? I guess there should be for example in ozone, but I'd like to see this discussed.

c) The question if the background conditions matter is partly related to a) because Sukhodolov et al. point out that they tried to do their simulations for historic 774/5 CE conditions. However, it is also prompted by the authors emphasizing already in the first sentence of the abstract that effects are "studied for the present-day climate and geomagnetic conditions". If this seems so important, I think some discussion is necessary about the implications of this assumption.

I'm listing a few further minor issues ordered by appearance in the text:

L10: "The results show a strong enhancement of NOx which causes a substantial decrease of ozone in the mesosphere and stratosphere, and a significant decrease of total ozone." I think the same sentence could have been written for a contemporary strong particle event due to the vagueness of the terms strong, substantial, and significant. I would like the authors to be more precise.

L23, figure headings, etc.: "additional ozone destruction". I guess that the authors want to point out that this ozone destruction is additional to the annual destruction by halogens, or only the anthropogenic destruction. However, as it isn't said anywhere and not quantified either how much the original destruction is, I find "additional" rather misleading.

L68: L* is not defined.

L72, and many other places: GMS, not GSM

L76, and several other places: "ionization production rate"? Rather "ionization rate" or "ion pair production rate", I guess.

L84: "ozone is the most important radiatively active gas for solar radiation". I think this depends on the metric.

L105: What are "basic equations"?

Fig.2 and others: I'd suggest to use color scales where not large parts of the signals are outside of the scale.

L171 "a kind of optimum" sounds very vague.

L176, and other places: Is "$NO_y$ input" or "injection" really an appropriate term? The additional $NO_y$ is produced in the atmosphere.

---

## Author Response (AR1)

**1  Answers to Reviewer 1**

First we want to thank Reviewer#1 for the encouraging review and especially for the tedious inspection of formatting the bibliography.

We changed all mentioned points according the suggestions.

**2  Answers to Reviewer 2**

First we want to thank Ref#2 for the valuable suggestions how the paper can be improved. Rev#2 raises three specific concerns:

1) Missing comparison with previous studies, here especially with Sukhodolov etal. (S2017):

In contrast to S2017, the current paper includes also a contribution from a geomagnetic storm to the solar particle forcing and evaluates its possible impact. The maximum ionization of the geomagnetic storm is here in the MLT region. NOy transported from the lower thermosphere into the middle atmosphere dominates the NOy input and can reach the stratosphere under favourable dynamical conditions during an elevated stratopause event. This cannot be studied with the setup of S2017, as for such a study a model with a top height reaching in the lower thermosphere is necessary (SOCOL has an upper boundary of about 80 km). To our knowledge, such an experiment has not been performed before. A sentence emphasizing this new aspect has been added in section 2 and the wording in the abstract has been changed slightly to make that clearer.

In the discussion section we now comment especially on the GMS results. In addition, we performed an GMS only experiment with 10x strength of the extreme scenario to clarify the role of GMSs. We only note this experiment in the text without showing a figure as the drawn conclusions are confirmed, i.e. that the impact of the GMS is small compared to the SPE.

We now also compare explicitly with the the results of S2017 (new subsection NOy, ozone).

A direct comparison with the results of S2017 as suggested by the reviewer seems to us not meaningful (besides the practical difficulties): S2017 focus on dynamical feedbacks. They look for strong ozone changes in the early winter in order to maximize radiative feedbacks and are finally searching for surface effects. Here, only an ensemble is capable to yield meaningful results. Our setup is more suited to analyze the direct chemical effects as we are using specified dynamics and can compare with a chemical reference run. As a result, our comparisons show higher impacts for composition but not the most probable impacts as S2017. With respect to total ozone, S2017 show in their Fig. S4 of the supplement the ozone loss in DU. We reach about the same initial ozone loss (with the date of the event in January first higher in SH), but lasting much longer in the NH compared to S2017. (S2017 give a maximum global decrease of total ozone of 8.5% which we cannot reproduce from the latitudinal distribution of the total ozone differences shown in their Fig. S4. Assuming a global mean of 300 DU we estimate a reduction of only between 1 - 2 % ). See also answer to 3).

2) Scaling property

This is a valid point which we now answer with a short subsection in the context of the ozone response. We added an experiment with a strength of 1/5 of the EXT and discuss the result.

3) Present day vs historic conditions:

This is related essentially with 1): a full comparison with a historic simulation like S2017 can only be accomplished by the model performing these simulations. From a chemical point of view, the flux of source gases like N2O into the stratosphere and their mean residence time determine the concentrations of tracers and the composition. Therefore, an analysis of the mean dynamical and chemical states under historic and present-day conditions would be necessary. There is no discussion in S2017 how the model results of the background state differ from present-day conditions. Even the simple fact that we expect the relative increase of NOy to be higher in S2017 just by the smaller pre-industrial concentration of N2O is only a guess as the N2O oxidation depends on the mean residence time and on ozone which depends itself also on the circulation.

Minor issues:

We followed the reviewer's suggestions, see the tracked changes and changed figures. Only with regard to 'NOy input' we would stay with this term as it describes the general case independent of local production or transport from the thermosphere.